# *SparseLLM*: Towards Global Pruning of Pre-trained Language Models

**Guangji Bai**[1]    **Yijiang Li**[2]    **Chen Ling**[1]    **Kibaek Kim**[2]    **Liang Zhao**[1,*]

[1] Emory University, Atlanta, GA, USA
[2] Argonne National Laboratory, Lemont, IL, USA
[*] Corresponding Author
{guangji.bai,chen.ling,liang.zhao}@emory.edu
{yijiang.li,kimk}@anl.gov

## Abstract

The transformative impact of large language models (LLMs) like LLaMA and GPT on natural language processing is countered by their prohibitive computational demands. Pruning has emerged as a pivotal compression strategy, introducing sparsity to enhance both memory and computational efficiency. Yet, traditional global pruning is impractical for LLMs due to scalability issues, while local pruning, despite its efficiency, leads to suboptimal solutions. Addressing these challenges, we propose *SparseLLM*, a novel framework that redefines the global pruning process into manageable, coordinated subproblems, allowing for resource-efficient optimization with global optimality. *SparseLLM*'s approach, which conceptualizes LLMs as a chain of modular functions and leverages auxiliary variables for problem decomposition, not only facilitates a pragmatic application on LLMs but also demonstrates significant performance improvements, particularly in high-sparsity regimes, surpassing current state-of-the-art methods. Our source code is publicly available at `https://github.com/BaiTheBest/SparseLLM`.

## 1 Introduction

Large language models (LLMs) [1, 2] have recently transformed the field of natural language processing (NLP) by delivering exceptional results across a variety of intricate language benchmarks [3, 4, 5]. Nonetheless, these models, with billions of parameters, generally necessitate significant computational resources. To make LLMs more accessible, extensive efforts have been devoted to model compression of LLMs [6, 7], including pruning, quantization, knowledge distillation, and low-rank factorization. *Pruning*, by introducing *sparsity*, jointly enhances memory and computational efficiency and offers unparalleled flexibility, seamlessly integrating with any LLMs, thus standing out as a highly effective and widely adopted compression strategy.

Model pruning has a long history [8] and has proven effective in applications related to vision and smaller language models [9]. However, conventional pruning techniques, which rely on global pruning and require loading the entire model into the same GPU [10, 11], become impractical for today's LLMs due to their vast size. Recently, several *local pruning* methods have been proposed for billion-scale LLMs. These methods compress each layer separately, and the overall compressed model is then obtained by "stitching together" the individually compressed layers. SparseGPT [12], an efficient unstructured pruning method for LLMs with hundreds of billions of parameters, achieves parameter reduction of up to 60% with minimal performance loss. Another approach, Wanda [13], introduces a novel pruning criterion that evaluates weights by considering both magnitude and related input activations. Despite its efficiency gains, local pruning only aims to minimize the local error for each specific layer under sparsity constraints, resulting in a *suboptimal* solution for the overall model.

38th Conference on Neural Information Processing Systems (NeurIPS 2024).

This is because local pruning *over-aligns* the intermediate layers' activations, leading to suboptimal performance, especially in high-sparsity regimes [11, 14].

To address these challenges and achieve global pruning with low memory consumption, we propose *SparseLLM* that decomposes the global pruning objective into multiple subproblems, each of which can be solved with low resources and coordinate to achieve the global pruning objective. More specifically, we first formulate LLMs as a composite function where the output of one module is the input of the next. Based on this formulation, we reformulate the global pruning goal into an equivalent form with auxiliary variables that facilitate its decomposition and coordination of the subproblems. Then we propose an alternating optimization algorithm to efficiently solve the subproblems, achieving computational resource efficiency and global optimality, due to the close-form solution of each subproblem. Empirically, we find that *SparseLLM* can consistently improve the performance of local pruning methods, particularly in high sparsity regimes (> 60%), where the perplexity can be significantly decreased by up to around 80% as compared to the state-of-the-art methods.

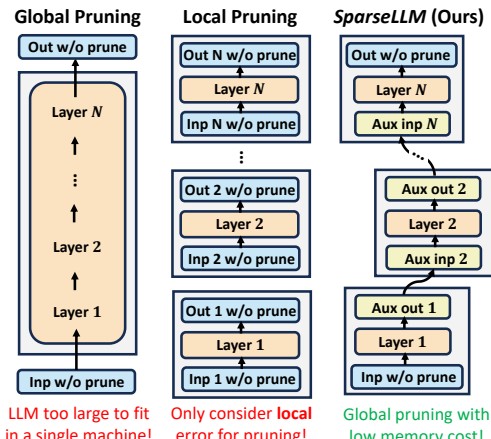

Figure 1: *SparseLLM* decomposes the global pruning of LLMs into manageable subproblems by leveraging the chain of modules and auxiliary variables while maintaining dependencies.

Furthermore, our SparseLLM framework can be readily applicable to enhance the performance of most existing local pruning solvers, such as SparseGPT and Wanda, with marginal additional computational overhead. This adaptability ensures that our framework can be seamlessly integrated into a wide range of LLMs and pruning methods, making it a versatile tool and useful baseline for future research exploiting the sparsity of LLMs.

## 2   Related work

*Pruning*, a pivotal concept in machine learning that introduces sparsity into neural networks, dates back to the 1980s [8]. It gained renewed attention in the late 2010s, especially for deep neural networks, under the banner of reducing inference costs [15]. LLM pruning techniques can broadly be categorized into *structured* and *unstructured* prunings.

Unstructured pruning [16, 17] looks at simplifying the complexity of LLMs by removing certain parameters *regardless* of the model's inherent structure. This approach typically involves setting a threshold to nullify parameters below it, leading to a model with a non-uniform sparse structure. SparseGPT [12], an efficient unstructured pruning method for LLMs with hundreds of billions of parameters, achieves up to 60% parameter reduction with minimal performance loss. A novel pruning criterion is introduced in Wanda [13], which evaluates weights by considering both magnitude and related input activations. This approach is beneficial in linear layers of LLMs, helping to identify and remove less significant weights. Tuli and Jha [18] proposed DynaTran, a dynamic inference scheme for pruning activations at runtime, supported by a specially designed ASIC architecture, AccelTran, to enhance transformer inference throughput.

On the other hand, structured pruning involves the selective removal of groups of weights, where "group" might mean blocks of weights, filters, attention heads, or other structures conducive to hardware acceleration. Ma et al. [19] introduced the LLM-Pruner, a framework designed for structured pruning of LLMs, which utilizes a combination of first-order data and Hessian information for effective importance estimation. This aids in identifying crucial groups for pruning. Li et al. [20] proposed LoSparse, a novel approach combining low-rank and sparse matrix approximations to balance pruning and expressive power. Tao et al. [21] extended this concept to pruning hidden dimensions in LLMs, including embedding layers and attention heads. ZipLM [22], a structured pruning method for LLMs, is proposed to optimize for compression and accuracy while considering specific hardware constraints. More recently, Xia et al introduced *LLM-shearing* [23], a structured pruning method that scales down LLaMA models by selectively pruning layers, heads, and dimensions.

This approach, combined with dynamic data batching, reduces pre-training compute costs while maintaining competitive performance, outperforming similar open-source models on key tasks.

Our work falls in the category of unstructured pruning of LLMs, where existing methods such as SparseGPT and Wanda only consider an *entirely local* pruning algorithm and suffer from *suboptimal* performance. We discuss the limitations and challenges of entirely local pruning in Sec. 3.

## 3 Background and notation

### 3.1 Global pruning

Given a pre-trained neural network $f$ with parameter $\mathbf{W}$ and inputs $\mathbf{X}$, global pruning aims to find a global sparsity mask $\mathbf{M}$ and possibly updated weights $\widehat{\mathbf{W}}$ to minimize the *global loss* $\mathcal{L}$ between the final outputs of the uncompressed and compressed model:

$$\min_{\mathbf{M},\widehat{\mathbf{W}}} \ \mathcal{L}\big(f(\mathbf{X}; \mathbf{M} \odot \widehat{\mathbf{W}}), f(\mathbf{X}; \mathbf{W})\big), \tag{1}$$

where $\odot$ denotes the *element-wise* multiplication. In addition to NP-hardness [24], however, a critical challenge in solving Eq. 1 is the huge memory cost, as one needs to store the entire model in a single GPU, rendering this method impractical for modern billion-scale LLMs.

### 3.2 Local pruning

Local pruning circumvents the memory issue mentioned above by dividing the full model compression into subproblems for each layer and constructing a *local loss* to measure the $\ell_2$-error between the outputs of the uncompressed and compressed layers. Hence, the local pruning can be formulated by

$$\min_{\mathbf{M}_\ell,\widehat{\mathbf{W}}_\ell} \|\mathbf{W}_\ell \cdot \mathbf{X}_\ell - (\mathbf{M}_\ell \odot \widehat{\mathbf{W}}_\ell) \cdot \mathbf{X}_\ell\|_2^2. \tag{2}$$

Although smaller than the global pruning, the local pruning still needs to optimize both the mask $\mathbf{M}_\ell$ and the remaining weights $\widehat{\mathbf{W}}_\ell$ and thus remains NP-hard. Therefore, exactly solving it for larger layers is unrealistic, leading all existing methods to resort to approximations.

**Mask selection & weight reconstruction.** A particularly popular approach is to separate the problem into *mask selection* and *weight reconstruction* [25, 26]. Concretely, this means first choosing a pruning mask $\mathbf{M}$ according to some salient criterion, like the weight magnitude [27], and then optimizing the remaining unpruned weights while keeping the mask unchanged. Importantly, once the mask is fixed, Eq. 2 turns into a *linear regression* problem that can be easily optimized.

**Existing solvers.** Early work [28] applied iterated linear regression to small networks. Recently, the AdaPrune approach [25] has shown good results for this problem on modern models via magnitude-based weight selection, followed by applying SGD steps to reconstruct the remaining weights. Follow-up works demonstrate that pruning accuracy can be further improved by removing the strict separation between mask selection and weight reconstruction. More recently, [12] developed SparseGPT, an efficient unstructured pruning method for LLMs with hundreds of billions of parameters, achieving up to 60% parameter reduction with minimal performance loss. [13] introduced a novel pruning criterion in Wanda, which evaluates weights by considering both magnitude and related input activations.

### 3.3 What is wrong with local pruning?

As shown in Eq. 2, local pruning focuses on minimizing the error for each specific layer $\ell$ subject to sparsity constraints. This results in a suboptimal solution with respect to the global pruning problem. While the primary goal of pruning is to ensure that the input and output of the pruned model align closely with those of the original models, the local pruning overly constrains the activations of all the intermediate layers between the two models, leading to performance degradation.

## 4 *SparseLLM*: Towards global pruning for LLMs

We present our proposed method SparseLLM that can address the drawbacks of existing pruning methods by achieving a global pruning with low memory consumption. SparseLLM decomposes the global pruning objective into many subproblems, each of which can be solved using low resources and can coordinate each other toward the global pruning objective. An overview of SparseLLM on the OPT and LlaMA configurations are shown in Figure 2.

### 4.1 Motivation

The development of SparseLLM is motivated by the observation: LLMs can be formulated as a composite function such that the output of one module is the input of the next. This allows us to reformulate the global pruning goal into its equivalent form with auxiliary variables that enable the decomposition into multiple subproblems, as detailed in Sec. 4.2. Then we develop a resource-efficient algorithm that achieves the alternating optimization of the subproblems with global optimality, thanks to the close-form solution of each subproblem, as illustrated in Sec. 4.3.

### 4.2 A unified formulation of pruning

In this section, we present the reformulation of the global pruning problem into an equivalent one by introducing auxiliary variables. This reformulation provides a more flexible form and enables the decomposition of the problem into many manageable subproblems.

The key idea behind our formulation is to decouple the densely parametric parts (linear layers) from non-parametric parts (activation function, self-attention, layer norm, etc) using a splitting technique. Rather than feeding the output of the dense linear layer $\mathbf{W}_\ell$ directly into the non-parametric and potentially nonlinear layer $\phi_\ell$, we store the output of layer $\ell$ in a new variable $\mathbf{z}_\ell = \mathbf{W}_\ell \boldsymbol{a}_{\ell-1}$ [1]. We also represent the output of the non-parametric layer as a vector of activations $\boldsymbol{a}_\ell = \phi_\ell(\mathbf{z}_\ell)$. We then solve the following problem:

$$
\begin{aligned}
\min_{\{\widehat{\mathbf{W}}_\ell\},\{\mathbf{M}_\ell\},\{\boldsymbol{a}_\ell\},\{\mathbf{z}_\ell\}} \quad & \mathcal{L}(\mathbf{z}_L, \mathbf{y}), \\
\text{s.t.} \quad & \mathbf{z}_\ell = (\mathbf{M}_\ell \odot \widehat{\mathbf{W}}_\ell)\boldsymbol{a}_{\ell-1}, \ \forall \, \ell \in [L], \\
& \boldsymbol{a}_\ell = \phi_\ell(\mathbf{z}_\ell), \ \forall \, \ell \in \Omega, \\
& \boldsymbol{a}_\ell, \mathbf{z}_\ell = \boldsymbol{a}_\ell^{pre}, \mathbf{z}_\ell^{pre}, \ \forall \, \ell \in [L-1]\backslash\Omega,
\end{aligned}
\tag{3}
$$

where $L$ represents the total number of dense (linear) layers and $[L] = \{1, 2, \cdots, L\}$. $[L-1]\backslash\Omega$ denotes the complement set of $\Omega$. We use $\boldsymbol{a}_\ell^{pre}$, $\mathbf{z}_\ell^{pre}$ to denote the corresponding intermediate variables' values of the original dense (i.e., *without* pruning) pre-trained model. $y$ denotes the ground-truth final output of the dense pre-trained model.

In our proposed formulation above, its unified nature lies in the interpretation and application of the set $\Omega$, which denotes the indices of layers subject to the pruning process. Intuitively, $\Omega$ measures how "global" the pruning is. The bigger the set of $\Omega$ is, the more layers are connected via the second constraint, and the pruning is more towards the global extreme, and vice versa. The generality and versatility of our formulation is illustrated in the following remark:

**Remark 4.1** (Generality and flexibility of Eq. 3)**.** *Given an LLM formulated as a composite function with dense layers $l \in \{1, 2, \ldots, L-1\}$, where $L$ is the total number of dense layers and $\Omega$ denotes the set of layers subject to the pruning process. Our formulation can seamlessly treat both global and local pruning as special cases under certain conditions. Specifically:*

- *When $\Omega = \{1, 2, \ldots, L-1\}$, solving our pruning formulation is equivalent to global pruning, accounting for inter-layer dependencies across the entire network.*

- *When $\Omega = \emptyset$, the formulation simplifies to local pruning, considering each layer independently (the last constraint dominates and "cuts" all layer dependencies with pre-trained values.)*

The ability to shift between these two extremes, and potentially any intermediate configurations, demonstrates the flexibility and comprehensiveness of our formulation. By adjusting $\Omega$, one can seamlessly transition from a global perspective to a local perspective. This flexibility not only caters to a wide range of pruning strategies but also provides a unified framework to compare and contrast the effectiveness of different pruning methods under a consistent mathematical lens.

### 4.3 Algorithm design

In this section, we introduce the algorithm design of *SparseLLM*, which alternatively optimizes the subproblems associated with the corresponding variables. This approach is resource-efficient and achieves global optimality, attributed to the closed-form solutions that each subproblem yields.

---

[1]For the sake of simplicity and clearer presentation, the bias term is omitted in the following equations where its exclusion does not lead to confusion.

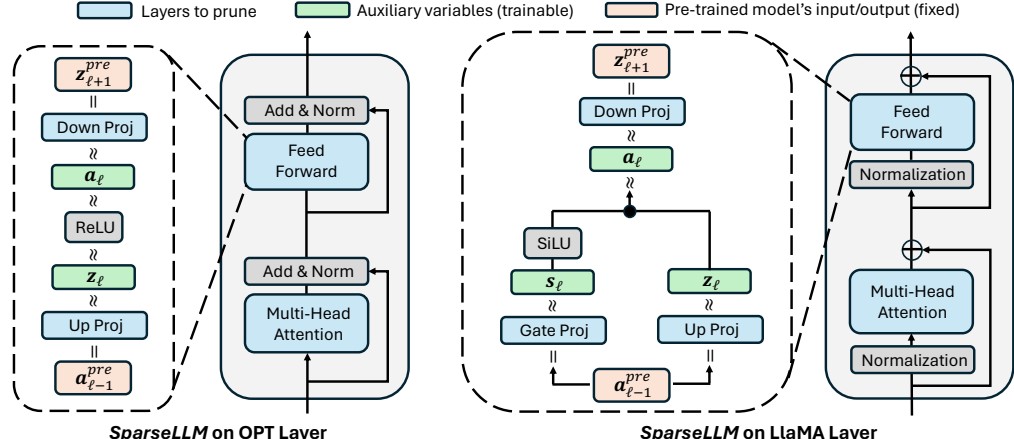

Figure 2: Illustration of *SparseLLM* on OPT and LlaMA. The auxiliary variables and soft constraints (i.e., ≈) allow *SparseLLM* to decompose the global pruning into manageable subproblems while maintaining the dependencies. Subproblems are *analytically* solvable and enjoy fast convergence.

The key idea of our algorithm lies behind the flexibility of $\Omega$ in our Eq. 3, as we want to find a better trade-off between completely global (memory bottleneck) and completely local (suboptimal performance) pruning. Naively applying SparseLLM to prune all layers globally is impractical. On the other hand, recent work shows that the feed-forward network (FFN) module in each decoder layer accounts for more than *two-thirds* of the total parameters in an LLM [29]. Therefore, our SparseLLM prioritizes the global pruning of the FFN module, while still adhering to a local pruning strategy for the multi-head attention (MHA) module (see Figure 2). This strategy strikes a balance between the computational feasibility of pruning large-scale models and the effectiveness of the pruning process, adhering to the limitations and practices of state-of-the-art LLM pruning frameworks.

Formally speaking, rather than trying to solve Eq. 3 directly, we first relax the constraints by adding an $\ell_2$-penalty function to the objective and attack the unconstrained problem:

$$\mathcal{L}(\mathbf{z}_L, \mathbf{y}) + \alpha \sum_{\ell \in [L]} \|\mathbf{z}_\ell - (\mathbf{M}_\ell \odot \widehat{\mathbf{W}}_\ell)\boldsymbol{a}_{\ell-1}\|_2^2 + \beta \sum_{\ell \in \Omega_{\text{FFN}}} \|\boldsymbol{a}_\ell - \phi_\ell(\mathbf{z}_\ell)\|_2^2, \qquad (4)$$

where $\alpha$, $\beta$ are hyperparameters for controlling the weight of each constraint. $\Omega_{\text{FFN}}$ denotes the set of indexes for the linear layers in the FFN module of each decoder layer, i.e., linear layers from the same FFN module are pruned globally. For simplicity, the superscript "pre" of $\boldsymbol{a}_\ell$ and $\mathbf{z}_\ell$ in the third constraint in Eq. 3 is omitted here, i.e., for $\ell \notin \Omega_{\text{FFN}}$ the $\boldsymbol{a}_\ell$ and $\mathbf{z}_\ell$ are fixed and equal to the pre-trained model's intermediate value in the second term of Eq. 4. In the following subsections, we illustrate how we approach the pruning of FFN and MHA modules, respectively.

### 4.3.1 *SparseLLM* on OPT models

For each decoder layer in a pre-trained LLM, our Eq. 4 instantly simplifies to globally pruning the corresponding FFN module within that decoder layer as:

$$\alpha\|\mathbf{z}_{\ell+1}^{pre} - (\mathbf{M}_{\ell+1} \odot \widehat{\mathbf{W}}_{\ell+1})\boldsymbol{a}_\ell\|_2^2 + \beta\|\boldsymbol{a}_\ell - \phi_\ell(\mathbf{z}_\ell)\|_2^2 + \alpha\|\mathbf{z}_\ell - (\mathbf{M}_\ell \odot \widehat{\mathbf{W}}_\ell)\boldsymbol{a}_{\ell-1}^{pre}\|_2^2, \qquad (5)$$

where layers $\ell$ and $\ell + 1$ correspond to the up-projection and down-projection linear layers.

In this work, we consider the *alternating* method to optimize our Eq. 5, i.e., optimize each variable while keeping the rest fixed. The careful and elaborate design of our Eq. 5 allows us to derive a *closed-form* solution to every subproblem as shown below.

**Pruning weight.** First consider optimizing Eq. 5 with respect to $\mathbf{M}_\ell$ and $\widehat{\mathbf{W}}_\ell$. For each linear layer $\ell$ in a FFN module, the optimal solution minimizes $\|\mathbf{z}_\ell - (\mathbf{M}_\ell \odot \widehat{\mathbf{W}}_\ell)\boldsymbol{a}_{\ell-1}\|_2^2$. To solve it, the first step is to decompose $\mathbf{z}_\ell$ to $\mathbf{W}_\ell \boldsymbol{a}_{\ell-1}$, where $\mathbf{W}_\ell = \mathbf{z}_\ell \boldsymbol{a}_{\ell-1}^\dagger$ († denotes the pseudo-inverse.) Plug decomposed $\mathbf{z}_\ell$ back in original loss and we get $\|\mathbf{W}_\ell \boldsymbol{a}_{\ell-1} - (\mathbf{M}_\ell \odot \widehat{\mathbf{W}}_\ell)\boldsymbol{a}_{\ell-1}\|_2^2$, which aligns with the pruning objective of Eq. 2 and can be analytically solved by existing pruning solver e.g., SparseGPT. The superscript of "*pre*" for $\boldsymbol{a}_{\ell-1}$ is omitted in this section for simpler notation.

**Updating activation.** Minimization for $\boldsymbol{a}_\ell$ is a simple least-squares problem similar to weight pruning. However, in this case, the matrix $\boldsymbol{a}_{\ell-1}$ appears in two penalty terms in Eq. 5, so we must minimize $\alpha\|\mathbf{z}_{\ell+1}^{pre} - (\mathbf{M}_{\ell+1} \odot \widehat{\mathbf{W}}_{\ell+1})\boldsymbol{a}_\ell\|_2^2 + \beta\|\boldsymbol{a}_\ell - \phi_\ell(\mathbf{z}_\ell)\|_2^2$ for $\boldsymbol{a}_\ell$, holding all other variables fixed. By following a very similar idea to Ridge regression, the new value of $\boldsymbol{a}_\ell$ is given by:

$$(\alpha\mathbf{W}_{\ell+1}^\intercal\mathbf{W}_{\ell+1} + \beta\mathbf{I})^{-1}(\alpha\mathbf{W}_{\ell+1}^\intercal\mathbf{z}_{\ell+1}^{pre} + \beta \cdot \mathrm{ReLU}(\mathbf{z}_\ell)), \tag{6}$$

where $\mathbf{W}_\ell$ denotes the updated weight matrix after pruning, i.e., $\mathbf{W}_\ell := \mathbf{M}_\ell \odot \widehat{\mathbf{W}}_\ell$.

**Updating output.** The update for $\mathbf{z}_\ell$ requires minimizing the following loss:

$$\beta\|\boldsymbol{a}_\ell - \mathrm{ReLU}(\mathbf{z}_\ell)\|_2^2 + \alpha\|\mathbf{z}_\ell - (\mathbf{M}_\ell \odot \widehat{\mathbf{W}}_\ell)\boldsymbol{a}_{\ell-1}^{pre}\|_2^2. \tag{7}$$

This problem is non-convex and non-quadratic (because of the non-linear function ReLU). Fortunately, because the ReLU function works entry-wise on its argument, the entries in $\mathbf{z}_\ell$ are de-coupled. Solving Eq. 7 is particularly easy for the case of ReLU, as it can be solved in closed form followed by a simple if-then logic. Specifically, one only needs to compute two solutions of a quadratic equation:

$$\mathbf{z}_\ell^{(1)} = (\mathbf{M}_\ell \odot \widehat{\mathbf{W}}_\ell)\boldsymbol{a}_{\ell-1}^{pre}, \quad \mathbf{z}_\ell^{(2)} = (\alpha + \beta)^{-1} \cdot (\beta\boldsymbol{a}_\ell + \alpha\mathbf{z}_\ell^{(1)}), \tag{8}$$

where the first solution corresponds to those entries of $\mathbf{z}_\ell$ that are negative (reduced to zero by ReLU), and the second solution corresponds to those entries of $\mathbf{z}_\ell$ that are non-negative.

### 4.3.2  *SparseLLM* on LlaMA models

In this section, we introduce how *SparseLLM* decomposes global pruning into subproblems and solves them iteratively on LlaMA model families. The model architecture of LlaMA can be found in Figure 2. Overall, *SparseLLM* operates similarly on both LlaMA and OPT models, with the main difference being that LlaMA includes an additional dense linear layer, known as the gate projection layer, and uses the SiLU activation function instead of ReLU.

**Pruning weight.** In this part, *SparseLLM* functions almost identically to its operation on OPTs.

**Updating activation $\mathbf{a}_\ell$.** Similarly, for updating $\boldsymbol{a}_\ell$, *SparseLLM* works nearly the same as on OPT. The minimization for $\boldsymbol{a}_\ell$ is a simple least-squares problem, akin to weight pruning. However, in this case, the matrix $\boldsymbol{a}_{\ell-1}$ appears in two penalty terms in Eq. 5, necessitating the minimization of:

$$\alpha\|\mathbf{z}_{\ell+1}^{pre} - (\mathbf{M}_{\ell+1} \odot \widehat{\mathbf{W}}_{\ell+1})\boldsymbol{a}_\ell\|_2^2 + \beta\|\boldsymbol{a}_\ell - \mathrm{SiLU}(\mathbf{s}_\ell) \odot \mathbf{z}_\ell\|_2^2, \tag{9}$$

for $\boldsymbol{a}_\ell$, with all other variables held fixed. Following a concept similar to Ridge regression, the updated value of $\boldsymbol{a}_\ell$ is:

$$(\alpha\mathbf{W}_{\ell+1}^\intercal\mathbf{W}_{\ell+1} + \beta\mathbf{I})^{-1}(\alpha\mathbf{W}_{\ell+1}^\intercal\mathbf{z}_{\ell+1}^{pre} + \beta \cdot \mathrm{SiLU}(\mathbf{s}_\ell) \odot \mathbf{z}_\ell), \tag{10}$$

where $\mathbf{W}_\ell$ denotes the updated weight matrix after pruning, i.e., $\mathbf{W}_\ell := \mathbf{M}_\ell \odot \widehat{\mathbf{W}}_\ell$.

**Updating output $\mathbf{z}_\ell$.** Updating $\mathbf{z}_\ell$ is somewhat simpler in LlaMA since the activation function applies over the gate projection layer. The update requires minimizing the loss:

$$\beta\|\boldsymbol{a}_\ell - \mathrm{SiLU}(\mathbf{s}_\ell) \odot \mathbf{z}_\ell\|_2^2 + \alpha\|\mathbf{z}_\ell - (\mathbf{M}_\ell \odot \widehat{\mathbf{W}}_\ell)\boldsymbol{a}_{\ell-1}^{pre}\|_2^2. \tag{11}$$

This problem is quadratic when solving for $\mathbf{z}_\ell$ with other variables fixed. Through mathematical manipulations, the analytical solution for $\mathbf{z}_\ell$ is found by solving a quadratic equation:

$$\mathbf{z}_\ell^* = \frac{(\mathbf{M}_\ell \odot \widehat{\mathbf{W}}_\ell)\boldsymbol{a}_{\ell-1}^{pre} + \mathrm{SiLU}(\mathbf{s}_\ell) \odot \boldsymbol{a}_\ell}{\mathrm{SiLU}(\mathbf{s}_\ell)^2 + \mathbf{1}}, \tag{12}$$

where the division is element-wise and $\mathbf{1}$ denotes the all-one matrix.

**Updating gate projection output $\mathbf{s}_\ell$.** Updating $\mathbf{s}_\ell$ involves minimizing:

$$\beta\|\boldsymbol{a}_\ell - \mathrm{SiLU}(\mathbf{s}_\ell) \odot \mathbf{z}_\ell\|_2^2 + \alpha\|\mathbf{s}_\ell - (\mathbf{M}_s \odot \widehat{\mathbf{W}}_s)\boldsymbol{a}_{\ell-1}^{pre}\|_2^2, \tag{13}$$

where $\mathbf{M}_s$ and $\widehat{\mathbf{W}}_s$ denote the mask and layer weights for the gate projection layer. This problem is non-convex and non-quadratic due to the non-linear SiLU function. However, since SiLU operates entry-wise, the entries in $\mathbf{s}_\ell$ are decoupled. Despite LlaMA lacking a simple closed-form solution as in OPT (which uses ReLU), the problem can still be solved quickly and analytically using a lookup table of pre-computed solutions, since each element in $\mathbf{s}_\ell$ depends on only three variables.

**Remark 4.2** (Global convergence of SparseLLM). *Consider the objective function given by Eq. 5, under the condition that the activation function $\phi$ is ReLU. Notice that (1) the objective function is convex with respect to each variable when all others are fixed, and (2) given that closed-form solutions exist for the subproblems in the alternating optimization scheme, the proposed algorithm resembles multiblock ADMM which has been shown to converge to in many applications.*

### 4.3.3 Pruning of MHAs

SparseLLM also prunes other linear layers besides those in FFNs. By following Eq. 4, for each linear layer out of FFN modules, the pruning objective simplifies to $\alpha\|\mathbf{z}_{\ell+1}^{pre} - (\mathbf{M}_{\ell+1} \odot \widehat{\mathbf{W}}_{\ell+1})\boldsymbol{a}_{\ell}^{pre}\|_2^2$, which is equivalent (with some simple math) to that of completely local pruning as shown in Eq. 2. Existing LLM pruning solvers such as SparseGPT and Wanda are applicable here.

### 4.4 Time complexity analyses

The proposed *SparseLLM* consists of three main steps, with the overall time complexity being the sum of the complexities of these steps. In the weights pruning step, the complexity is dominated by the pseudo-inverse computation of matrix $\boldsymbol{a}_\ell$ (dimensions $n \times h$), which is $O(nh^2)$. Using SparseGPT as the solver, the exact pruning step has a complexity of $O(h^3)$. The second step, updating activations, involves matrix inversion of the weight matrix $\mathbf{W}_\ell$ (size $h \times h$) with a complexity of $O(h^3)$. The third step, updating outputs, has a lower complexity. Thus, the overall algorithm complexity is bounded by $O(h^3)$, therefore making our method's per-epoch time complexity comparable to SparseGPT.

## 5 Experiments

**Experiments setup.** We implemented *SparseLLM* in PyTorch [30] and use the HuggingFace Transformers library [31] for handling models and datasets. All pruning experiments are conducted on NVIDIA A100 GPUs. For calibration data, we follow [12] and use 128 2048-token segments, randomly chosen from the first shard of the C4 [32] dataset. This represents generic text data crawled from the internet and ensures our experiments are zero-shot as no task-specific data is seen during pruning. *We followed existing work [12, 13] and pruned all linear layers (in FFN and MHA) to the target sparsity.*

**Models, datasets & evaluation.** We consider the OPT model family [33] and LlaMA-2 model family [1] in our experiments as well as the most recent LlaMA-3 model. We show results on different sizes of models to provide a broader picture for the performances of *SparseLLM*. In terms of metrics, we mainly focus on perplexity, which is known to be a challenging and stable metric that is well-suited for evaluating the accuracy of compression methods [34, 35]. We consider the test sets of raw-WikiText2 [36] (WT2) and PTB [37] as well as a subset of the C4 validation data, all popular benchmarks in LLM compression literature [34, 38, 12, 13]. For additional interpretability, we also provide zero-shot accuracy results following the same setup of [13], which is based on the popular EleutherAI-eval harness [39].

**Comparison methods.** We compare against three baselines, magnitude pruning [27] applied locally, and two other state-of-the-art local pruning methods, SparseGPT [12] and Wanda [13].

### 5.1 Results and analyses

**Pruning vs. model sizes.** We begin by exploring the pruning capabilities of *SparseLLM* across various model sizes in comparison to baseline methods. For each model, we consider unstructured sparsity ranging from 70% to 90% with a 10% increment, as well as a 3:4 semi-structured sparsity. The 3:4 semi-structured sparsity is inspired by our preliminary results that suggest good performance *SparseLLM* at high sparsity regimes. However, note that two of our baselines, Magnitude and Wanda, are unable to be configured to this sparsity out-of-box. We conduct a sensitivity study on the calibration sample sizes (see Appendix A.3) and use calibration sample sizes between 32 and 64 for all experiments. Moreover, we prune the first 50% of the Transformer decoder layers in each model to achieve a balance between the computation resources and the performances. Detailed results can be found in Table 1 and Table 2 as well as Table 8 in Appendix A.5. Note that in Table 2 for LlaMA-3 model, we only compare SparseGPT to the proposed *SparseLLM*. The perplexity results of the dense models are reported next to the names of the models.

From the tables, it shows a general trend of increasing perplexity with increasing sparsity. Moreover, we observe a trend of decreasing perplexity for SparseGPT and *SparseLLM* at the same sparsity

Table 1: Perplexity of OPT models for sparsity $\geq 70\%$; the lower the perplexity, the better.

| OPT-1.3b (WikiText2 (WT2): 14.62; PTB: 20.29; C4: 16.07) | | | | | | | | | | | |
|---|---|---|---|---|---|---|---|---|---|---|---|
| Sparsity | 70% | | | 80% | | | 90% | | | 3:4 | | |
| Dataset | WT2 | PTB | C4 | WT2 | PTB | C4 | WT2 | PTB | C4 | WT2 | PTB | C4 |
| Magnitude | 6420.80 | 4828.13 | 3435.99 | 9998.71 | 1.1e4 | 5347.89 | 8209.13 | 1.0e4 | 4917.02 | - | - | - |
| Wanda | 21.56 | 34.77 | 25.78 | 142.20 | 146.76 | 142.24 | 5692.65 | 4751.69 | 4501.73 | - | - | - |
| SparseGPT | 18.04 | 28.19 | 21.45 | 69.67 | 93.36 | 60.83 | 2596.70 | 2361.86 | 1363.08 | 252.81 | 238.41 | 146.21 |
| SparseLLM | 17.82 | 27.72 | 20.99 | 58.92 | 85.33 | 58.36 | 1350.31 | 1192.36 | 655.76 | 128.83 | 144.48 | 106.01 |

| OPT-2.7b (WikiText2 (WT2): 12.47; PTB: 17.97; C4: 14.32) | | | | | | | | | | | |
|---|---|---|---|---|---|---|---|---|---|---|---|---|
| Sparsity | 70% | | | 80% | | | 90% | | | 3:4 | | |
| Dataset | WT2 | PTB | C4 | WT2 | PTB | C4 | WT2 | PTB | C4 | WT2 | PTB | C4 |
| Magnitude | 1691.74 | 1237.08 | 1415.02 | 1.0e4 | 7916.69 | 6050.07 | 7.9e5 | 5.3e5 | 4.7e5 | - | - | - |
| Wanda | 88.61 | 140.09 | 90.06 | 6140.81 | 4746.96 | 5678.66 | 3.0e4 | 3.5e4 | 2.4e4 | - | - | - |
| SparseGPT | 13.79 | 21.18 | 16.18 | 24.32 | 37.82 | 25.92 | 2662.74 | 2285.01 | 1776.08 | 91.02 | 91.79 | 64.95 |
| SparseLLM | 13.82 | 21.07 | 16.14 | 23.87 | 37.09 | 24.90 | 1200.12 | 759.11 | 527.70 | 56.90 | 77.14 | 52.77 |

| OPT-13b (WikiText2 (WT2): 10.13; PTB: 14.52; C4: 12.06) | | | | | | | | | | | |
|---|---|---|---|---|---|---|---|---|---|---|---|---|
| Sparsity | 70% | | | 80% | | | 85% | | | 3:4 | | |
| Dataset | WT2 | PTB | C4 | WT2 | PTB | C4 | WT2 | PTB | C4 | WT2 | PTB | C4 |
| Magnitude | 9037.12 | 7734.58 | 5909.47 | 1.1e4 | 9140.88 | 6340.22 | 1.3e4 | 1.3e4 | 9087.50 | - | - | - |
| Wanda | 30.94 | 39.26 | 33.31 | 4216.04 | 2894.77 | 2450.57 | 1.1e4 | 1.1e4 | 7244.96 | - | - | - |
| SparseGPT | 10.89 | 16.35 | 13.39 | 21.42 | 33.62 | 21.01 | 8408.03 | 6380.30 | 3416.23 | 4715.16 | 7454.37 | 2.11e4 |
| SparseLLM | 10.96 | 16.57 | 13.38 | 19.07 | 28.77 | 19.29 | 2052.27 | 1536.51 | 538.61 | 289.17 | 687.48 | 677.13 |

| OPT-30b (WikiText2 (WT2): 9.56; PTB: 14.04; C4: 11.45) | | | | | | | | | | | |
|---|---|---|---|---|---|---|---|---|---|---|---|---|
| Sparsity | 70% | | | 80% | | | 90% | | | 3:4 | | |
| Dataset | WT2 | PTB | C4 | WT2 | PTB | C4 | WT2 | PTB | C4 | WT2 | PTB | C4 |
| Magnitude | 8691.40 | 4769.89 | 4732.66 | 8941.81 | 5292.98 | 5092.26 | 3.8e7 | 3.0e7 | 1.4e7 | - | - | - |
| Wanda | 7766.61 | 5547.45 | 5741.74 | 8770.33 | 6020.70 | 7132.20 | 6354.15 | 4296.37 | 4654.27 | - | - | - |
| SparseGPT | 9.58 | 14.41 | 11.93 | 16.49 | 22.01 | 17.67 | 5747.87 | 5169.50 | 3555.24 | 441.35 | 464.73 | 209.44 |
| SparseLLM | 9.56 | 14.40 | 11.94 | 15.61 | 19.64 | 16.61 | 3050.63 | 2712.39 | 1758.63 | 51.28 | 73.61 | 37.99 |

| OPT-66b (WikiText2 (WT2): 9.34; PTB: 13.36; C4: 10.99) | | | | | | | | | | | |
|---|---|---|---|---|---|---|---|---|---|---|---|---|
| Sparsity | 70% | | | 80% | | | 90% | | | 3:4 | | |
| Dataset | WT2 | PTB | C4 | WT2 | PTB | C4 | WT2 | PTB | C4 | WT2 | PTB | C4 |
| Magnitude | OOM | OOM | OOM | OOM | OOM | OOM | OOM | OOM | OOM | - | - | - |
| Wanda | OOM | OOM | OOM | OOM | OOM | OOM | OOM | OOM | OOM | - | - | - |
| SparseGPT | 9.45 | 13.64 | 11.37 | 28.27 | 57.41 | 26.26 | 7803.10 | 6594.88 | 4433.35 | 6594.37 | 6329.59 | 3799.87 |
| SparseLLM | 9.37 | 13.66 | 11.37 | 16.45 | 21.00 | 17.70 | 7504.17 | 5644.65 | 3683.91 | 4641.8 | 5296.93 | 1618.43 |

Table 2: Perplexity of LlaMA models for sparsity $\geq 70\%$; the lower the perplexity, the better.

| LlaMA-2 7b (WikiText2 (WT2): 5.47; PTB: 37.91; C4: 7.26) | | | | | | | | | | | |
|---|---|---|---|---|---|---|---|---|---|---|---|---|
| Sparsity | 70% | | | 80% | | | 90% | | | 3:4 | | |
| Dataset | WT2 | PTB | C4 | WT2 | PTB | C4 | WT2 | PTB | C4 | WT2 | PTB | C4 |
| Magnitude | 1058.00 | 544.43 | 889.46 | 6380.27 | NaN | 4162.92 | 9498.91 | 1.02e4 | 7539.65 | - | - | - |
| Wanda | 2644.22 | 4040.95 | 1630.09 | 1814.01 | 3376.35 | 1124.26 | 5206.93 | 4607.30 | 2780.45 | - | - | - |
| SparseGPT | 15.98 | 302.15 | 18.58 | 53.20 | 803.02 | 52.57 | 344.97 | 2503.82 | 279.77 | 68.28 | 784.79 | 60.45 |
| SparseLLM | 16.15 | 274.35 | 18.23 | 49.96 | 664.39 | 47.39 | 225.23 | 2233.52 | 181.56 | 64.17 | 667.27 | 54.56 |

| LlaMA-2 13b (WikiText2 (WT2): 4.88; PTB: 50.94; C4: 6.73) | | | | | | | | | | | |
|---|---|---|---|---|---|---|---|---|---|---|---|---|
| Sparsity | 70% | | | 80% | | | 90% | | | 3:4 | | |
| Dataset | WT2 | PTB | C4 | WT2 | PTB | C4 | WT2 | PTB | C4 | WT2 | PTB | C4 |
| Magnitude | 30.34 | 2317.39 | 28.48 | 4133.98 | 4706.65 | 4112.69 | 5580.71 | 5514.22 | 5090.63 | - | - | - |
| Wanda | 23.42 | 502.53 | 32.65 | 295.29 | 2340.13 | 261.15 | 3003.49 | 3804.69 | 1738.73 | - | - | - |
| SparseGPT | 12.98 | 267.63 | 15.95 | 45.59 | 550.59 | 45.20 | 825.99 | 1410.46 | 673.33 | 63.48 | 660.70 | 56.29 |
| SparseLLM | 12.95 | 277.76 | 15.77 | 36.36 | 578.35 | 38.63 | 646.15 | 1078.94 | 466.98 | 53.71 | 632.11 | 50.40 |

| LlaMA-3 8b (WikiText2 (WT2): 6.14; PTB: 11.18; C4: 9.45) | | | | | | | | | | | |
|---|---|---|---|---|---|---|---|---|---|---|---|---|
| Sparsity | 70% | | | 80% | | | 90% | | | 3:4 | | |
| Dataset | WT2 | PTB | C4 | WT2 | PTB | C4 | WT2 | PTB | C4 | WT2 | PTB | C4 |
| SparseGPT | 22.37 | 36.56 | 30.53 | 72.87 | 113.95 | 79.86 | 214.68 | 261.18 | 198.34 | 96.75 | 107.52 | 102.11 |
| *SparseLLM* | 20.98 | 33.78 | 28.94 | 57.83 | 85.98 | 72.18 | 197.47 | 241.68 | 181.69 | 76.33 | 99.54 | 93.68 |

Table 3: Perplexity of 2:4 sparsity; the lower the perplexity, the better.

| Dataset | OPT-1.3b | | | OPT-2.7b | | | OPT-6.7b | | | OPT-13b | | |
| | WT2 | PTB | C4 | WT2 | PTB | C4 | WT2 | PTB | C4 | WT2 | PTB | C4 |
|---|---|---|---|---|---|---|---|---|---|---|---|---|
| Magnitude | 96.68 | 133.92 | 48.08 | 272.34 | 308.55 | 267.70 | 64.11 | 92.23 | 82.67 | 67.07 | 110.77 | 52.61 |
| Wanda | 15.63 | 24.04 | 18.23 | 13.66 | 21.67 | 16.10 | 11.86 | 18.54 | 14.77 | 10.33 | 15.35 | 12.54 |
| SparseGPT | 15.11 | 23.71 | 17.88 | 12.62 | 19.28 | 15.12 | 11.30 | 16.90 | 13.51 | 10.20 | 15.14 | 12.48 |
| *SparseLLM* | 14.97 | 23.40 | 17.67 | 12.62 | 19.28 | 15.12 | 11.07 | 16.73 | 13.42 | 10.20 | 15.14 | 12.41 |

with increasing model sizes. However, such a trend is not obvious for Magnitude and Wanda. We also observe that SparseGPT and *SparseLLM* consistently outperform Magnitude and Wanda by a significant margin. For smaller sparsity, *SparseLLM* achieves comparable perplexity to SparseGPT. As we increase the sparsity, *SparseLLM* starts to demonstrate noticeable improvements over SparseGPT. In numerous instances for the OPT model family, *SparseLLM* achieves perplexity reductions of more than 50% compared to SparseGPT. We also see that performance improvements from *SparseLLM* over SparseGPT are more significant for the OPT model family than the LlaMA-2 model family.

We provide additional set of perplexity results for a 2:4 semi-structured sparsity for a few OPT models in Table 3. We see that *SparseLLM* and SparseGPT generally outperform Magnitude and Wanda while *SparseLLM* has comparable if not slightly better performances compared to SparseGPT with the 2:4 semi-structured sparsity. Note that a 2:4 semi-structure sparsity is considered to be in low sparsity regime.

**Zero-shot experiments.** To further conclude the evaluations and discussions, we show results for several zero-shot tasks in Table 4 and Table 5 as well as Table 9 in Appendix A.5, comparing SparseGPT and *SparseLLM*. These evaluations are known to be relatively noisy [40], but more interpretable. We also report the results for zero-shot tasks from the dense models in the "Dense" row. We see that the accuracy of both methods decreases with increasing sparsity, which is expected, as more parameters are pruned. A similar trend of increasing accuracy

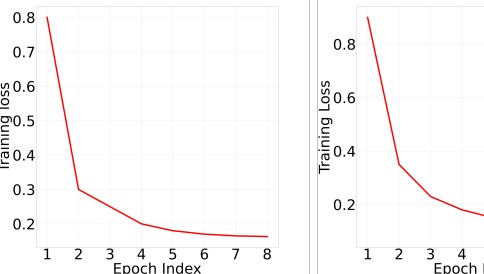

Figure 3: **Fast convergence** of *SparseLLM*. Training loss per epoch for pruning layer 3 of OPT-125m at 80% sparsity (**Left**) and layer 6 of LlaMA-2 13b at 70% sparsity (**Right**).

with increasing model size is observed too. Across all the tasks, OBQA and ARC-c remain the most challenging ones as the accuracy for both methods is 30% or below 30% while both methods perform well for BoolQ, RTE, WinoGrande, and ARC-e. In general, *SparseLLM* is able to achieve higher accuracy in the majority of tasks across the models of different sizes in both OPT and LlaMA-2 model families.

**Training loss vs. epochs in *SparseLLM*.** Figure 3 illustrates the change in training loss over epochs for *SparseLLM*, with the training loss plotted on a scale of $10^3$ for clarity. We observe that the training loss decreases rapidly during the initial epochs, highlighting the efficiency of *SparseLLM* in achieving effective global pruning within a short period. This rapid convergence is largely due to the closed-form solutions employed by *SparseLLM* for various subproblems, which streamline the pruning process and ensure optimal layer-wise pruning without extensive iterative computations. These analytical solutions enable *SparseLLM* to perform precise pruning operations quickly, making it a powerful tool for optimizing large-scale models like LlaMA, significantly reducing model size while maintaining high accuracy.

## 6 Conclusion

Our work presents *SparseLLM*, a cutting-edge framework poised to redefine the compression of LLMs through sparsity. By adeptly circumventing the scalability issues of global pruning and optimizing the local suboptimality of existing methods, *SparseLLM* stands as a significant advancement in the field. Our empirical results affirm its efficacy, particularly in high-sparsity environments. It achieves a notable reduction in perplexity, thereby setting a new precedent for model compression. The versatility and minimal computational overhead of *SparseLLM* complement its integration

Table 4: Accuracy (%) of zero-shot tasks for OPT models; the higher the accuracy, the better.

| | | | | | OPT-13b | | | | |
|---|---|---|---|---|---|---|---|---|---|
| Sparsity | Method | BoolQ | RTE | HellaSwag | WinoGrande | ARC-e | ARC-c | OBQA | Mean |
| Dense | | 65.87 | 57.76 | 52.44 | 66.02 | 67.82 | 33.46 | 28.62 | 53.14 |
| 70% | SparseGPT | 63.03 | 54.87 | 50.89 | 65.43 | 67.47 | 32.85 | 26.40 | 51.56 |
| | SparseLLM | 63.85 | 55.23 | 50.73 | 65.67 | 66.46 | 31.83 | 27.20 | 51.57 |
| 80% | SparseGPT | 59.72 | 52.35 | 46.82 | 61.48 | 62.50 | 31.23 | 21.80 | 47.99 |
| | SparseLLM | 60.89 | 53.07 | 46.19 | 62.12 | 62.21 | 30.38 | 23.00 | 48.27 |
| 90% | SparseGPT | 47.49 | 52.71 | 33.17 | 51.54 | 39.98 | 21.33 | 17.80 | 37.72 |
| | SparseLLM | 53.43 | 52.71 | 38.19 | 52.96 | 46.68 | 25.26 | 17.40 | 40.95 |
| 3:4 | SparseGPT | 47.55 | 53.43 | 31.30 | 50.20 | 37.63 | 22.53 | 17.60 | 37.18 |
| | SparseLLM | 51.13 | 52.35 | 38.51 | 55.96 | 49.24 | 24.83 | 21.40 | 41.92 |
| | | | | | OPT-30b | | | | |
| Sparsity | Method | BoolQ | RTE | HellaSwag | WinoGrande | ARC-e | ARC-c | OBQA | Mean |
| Dense | | 70.46 | 61.82 | 54.27 | 69.02 | 70.47 | 35.49 | 30.20 | 55.96 |
| 70% | SparseGPT | 68.78 | 58.48 | 53.83 | 67.64 | 69.15 | 34.30 | 29.60 | 54.54 |
| | SparseLLM | 69.11 | 61.73 | 53.97 | 68.43 | 69.78 | 34.73 | 29.80 | 55.36 |
| 80% | SparseGPT | 64.86 | 60.65 | 49.73 | 61.40 | 61.91 | 31.74 | 24.20 | 50.64 |
| | SparseLLM | 65.41 | 59.57 | 50.65 | 61.96 | 62.71 | 32.25 | 26.50 | 51.29 |
| 90% | SparseGPT | 37.83 | 53.79 | 25.96 | 49.88 | 26.47 | 20.22 | 12.60 | 32.39 |
| | SparseLLM | 43.55 | 52.35 | 26.32 | 50.04 | 27.31 | 20.56 | 14.00 | 33.45 |
| 3:4 | SparseGPT | 55.81 | 51.26 | 33.64 | 54.54 | 42.05 | 21.33 | 21.00 | 39.95 |
| | SparseLLM | 60.83 | 54.15 | 39.35 | 55.41 | 45.24 | 24.06 | 22.20 | 43.03 |

Table 5: Accuracy (%) of zero-shot tasks for LlaMA models; the higher the accuracy, the better.

| | | | | | LlaMA-2 7b | | | | |
|---|---|---|---|---|---|---|---|---|---|
| Sparsity | Method | BoolQ | RTE | HellaSwag | WinoGrande | ARC-e | ARC-c | OBQA | Mean |
| Dense | | 75.05 | 66.43 | 56.92 | 69.93 | 75.34 | 41.89 | 34.40 | 59.99 |
| 70% | SparseGPT | 68.26 | 57.04 | 39.67 | 59.04 | 60.9 | 28.58 | 20.60 | 47.73 |
| | SparseLLM | 67.61 | 57.31 | 40.12 | 61.39 | 59.39 | 28.76 | 21.40 | 48.13 |
| 80% | SparseGPT | 59.36 | 52.71 | 28.83 | 48.7 | 34.22 | 18.34 | 14.40 | 36.65 |
| | SparseLLM | 60.12 | 53.07 | 28.62 | 50.59 | 34.55 | 18.69 | 14.30 | 37.13 |
| 90% | SparseGPT | 39.02 | 52.34 | 26.66 | 47.80 | 28.32 | 17.37 | 12.40 | 31.99 |
| | SparseLLM | 39.45 | 52.71 | 26.79 | 51.17 | 28.32 | 19.52 | 12.50 | 32.92 |
| 3:4 | SparseGPT | 53.94 | 54.15 | 28.09 | 49.17 | 31.57 | 17.41 | 14.80 | 35.59 |
| | SparseLLM | 57.34 | 53.43 | 28.26 | 48.86 | 32.45 | 18.17 | 14.4 | 36.13 |
| | | | | | LlaMA-2 13b | | | | |
| Sparsity | Method | BoolQ | RTE | HellaSwag | WinoGrande | ARC-e | ARC-c | OBQA | Mean |
| Dense | | 77.89 | 70.40 | 59.94 | 72.77 | 77.40 | 46.50 | 33.20 | 62.59 |
| 70% | SparseGPT | 70.03 | 53.43 | 42.20 | 66.54 | 64.94 | 31.66 | 25.40 | 50.60 |
| | SparseLLM | 69.87 | 54.15 | 42.50 | 68.64 | 64.97 | 31.40 | 25.80 | 51.05 |
| 80% | SparseGPT | 62.69 | 52.71 | 28.94 | 50.91 | 36.24 | 18.17 | 14.00 | 37.67 |
| | SparseLLM | 64.39 | 52.86 | 29.19 | 51.46 | 35.69 | 18.77 | 14.20 | 38.08 |
| 90% | SparseGPT | 50.21 | 51.35 | 26.71 | 49.14 | 26.68 | 19.71 | 13.2 | 33.86 |
| | SparseLLM | 55.35 | 52.05 | 26.89 | 51.34 | 27.35 | 19.62 | 14.20 | 35.26 |
| 3:4 | SparseGPT | 61.28 | 53.71 | 28.40 | 47.99 | 33.21 | 18.26 | 14.00 | 36.69 |
| | SparseLLM | 61.71 | 55.71 | 28.56 | 51.62 | 32.11 | 18.49 | 13.8 | 37.43 |

with current pruning technologies, underscoring its potential as a universal tool for enhancing the performance and accessibility of LLMs.

## Acknowledgments and Disclosure of Funding

This work was supported by the National Science Foundation (NSF) Grant No.1755850, No.1841520, No.1942594, No.2403312, No.2007716, No.2007976, No.1907805. This work was supported by the U.S. Department of Energy, Office of Science, Advanced Scientific Computing Research, under Contract DE-AC02-06CH11357. This research used resources of the Argonne Leadership Computing Facility at Argonne National Laboratory, which is supported by the Office of Science of the U.S. Department of Energy under contract DE-AC02-06CH11357.

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

# A Appendix

This section includes supplemental materials (pseudo-code, additional experiments, and plots).

## A.1 Pseudo-code of *SparseLLM*

---

**Algorithm 1** *SparseLLM* Pruning of OPT Models.

---

**Input:** An OPT decoder layer containing FFN and MHA modules. FFN's up-scaling linear layer pre-trained weight matrix $\mathbf{W}_\ell$, FFN's down-scaling linear layer pre-trained weight matrix $\mathbf{W}_{\ell+1}$, input of the up-scaling linear layer $\boldsymbol{a}_{\ell-1}^{pre}$, output of the down-scaling linear layer $\mathbf{z}_{\ell+1}^{pre}$, target sparsity $\rho$, constraint weight hyperparameters $\alpha, \beta$.

1   SparseLLM on FFN():

2      Initialize $\mathbf{z}_\ell = \mathbf{z}_\ell^{pre}, \boldsymbol{a}_\ell = \boldsymbol{a}_\ell^{pre}$               ▷ Initialize slack variables

      Pre-compute and cache $\boldsymbol{a}_{\ell-1}^\dagger$ = pseudo-inverse($\boldsymbol{a}_{\ell-1}^{pre}$)

      **for** *step* $i = 1, \cdots, K$ **do**

3         $\mathbf{W}_\ell = \mathbf{z}_\ell \boldsymbol{a}_{\ell-1}^\dagger, \ \ \mathbf{W}_{\ell+1} = \mathbf{z}_{\ell+1} \boldsymbol{a}_\ell^\dagger$

         $\mathbf{M}_\ell, \widehat{\mathbf{W}}_\ell = \arg\min \|\mathbf{W}_\ell \boldsymbol{a}_{\ell-1}^{pre} - (\mathbf{M}_\ell \odot \widehat{\mathbf{W}}_\ell) \boldsymbol{a}_{\ell-1}^{pre}\|_2^2$      ▷ Prune layer $\ell$ by SparseGPT solver

         $\mathbf{M}_{\ell+1}, \widehat{\mathbf{W}}_{\ell+1} = \arg\min \|\mathbf{W}_{\ell+1} \boldsymbol{a}_\ell - (\mathbf{M}_{\ell+1} \odot \widehat{\mathbf{W}}_{\ell+1}) \boldsymbol{a}_\ell\|_2^2$    ▷ Prune layer $\ell + 1$ by SparseGPT solver

         $\mathbf{W}_{\ell+1} = \mathbf{M}_{\ell+1} \odot \widehat{\mathbf{W}}_{\ell+1}, \ \ \mathbf{W}_\ell = \mathbf{M}_\ell \odot \widehat{\mathbf{W}}_\ell$

         $\boldsymbol{a}_\ell = (\alpha \mathbf{W}_{\ell+1}^\intercal \mathbf{W}_{\ell+1} + \beta \mathbf{I})^{-1} (\alpha \mathbf{W}_{\ell+1}^\intercal \mathbf{z}_{\ell+1}^{pre} + \beta \phi_\ell(\mathbf{z}_\ell))$         ▷ Update activations

         $\mathbf{z}_\ell^{(1)} = \mathbf{W}_\ell \boldsymbol{a}_{\ell-1}^{pre}, \ \ \mathbf{z}_\ell^{(2)} = (\alpha + \beta)^{-1} \cdot (\beta \boldsymbol{a}_\ell + \alpha \mathbf{z}_\ell^{(1)}),$

         **for** $j = 1, \cdots, n$ *in parallel* **do**

4            **if** $(\mathbf{z}_\ell)_j < 0$ **then**

5              $(\mathbf{z}_\ell)_j = (\mathbf{z}_\ell^{(1)})_j$                 ▷ Update outputs

6            **else**

7              $(\mathbf{z}_\ell)_j = (\mathbf{z}_\ell^{(2)})_j$                 ▷ Update outputs

8      **return** $\mathbf{W}_\ell, \mathbf{W}_{\ell+1}$

9   SparseLLM on MHA():

10      **for** *each linear layer $\ell$ in MHA module* **do**

11         Fix $\mathbf{z}_\ell = \mathbf{z}_\ell^{pre}, \boldsymbol{a}_\ell = \boldsymbol{a}_\ell^{pre}$           ▷ Fix intermediate variables

         $\mathbf{M}_\ell, \widehat{\mathbf{W}}_\ell = \arg\min \|\mathbf{W}_\ell \boldsymbol{a}_{\ell-1}^{pre} - (\mathbf{M}_\ell \odot \widehat{\mathbf{W}}_\ell) \boldsymbol{a}_{\ell-1}^{pre}\|_2^2$      ▷ Prune layer $\ell$ by SparseGPT solver

12      **return** $\{\mathbf{W}_\ell\}$ for all linear layers in MHA

---

The *SparseLLM* algorithm presented in Algorithm 1 demonstrates how *SparseLLM* works on an OPT decoder layer. The key inputs to the algorithm include the pre-trained weight matrices for both the up-scaling and down-scaling linear layers of the FFN, along with a set of hyperparameters and a sparsity constraint. The goal of *SparseLLM* is to achieve a targeted level of sparsity in the linear layers without significantly compromising the model's performance.

Initiating with the pre-trained weights, *SparseLLM* employs a series of pruning and activation update steps across $K$ iterations. In each iteration, it solves optimization problems to prune the current and subsequent layer weights, followed by updating the activation variables. The utilization of SparseGPT solvers for pruning and the strategic update of activations ensures that the pruned network approximates the original network's behavior as closely as possible. The final output of the algorithm is a pair of pruned weight matrices for the consecutive layers, which are expected to deliver comparable or improved performance with a reduced number of parameters.

## A.2 Two-layer Demo on the Details behind our Global Pruning

Figure 4 illustrates the *SparseLLM* pruning method compared to conventional global pruning and local pruning, using a two-layer neural network as an abstraction for simplicity. The figure is divided into three main parts:

On the left, conventional global pruning is depicted. This method applies a global mask to the entire network, resulting in significant memory costs due to poor scalability. Both functions $f_1$ and $f_2$ are pruned using the same mask across all layers, leading to high memory usage.

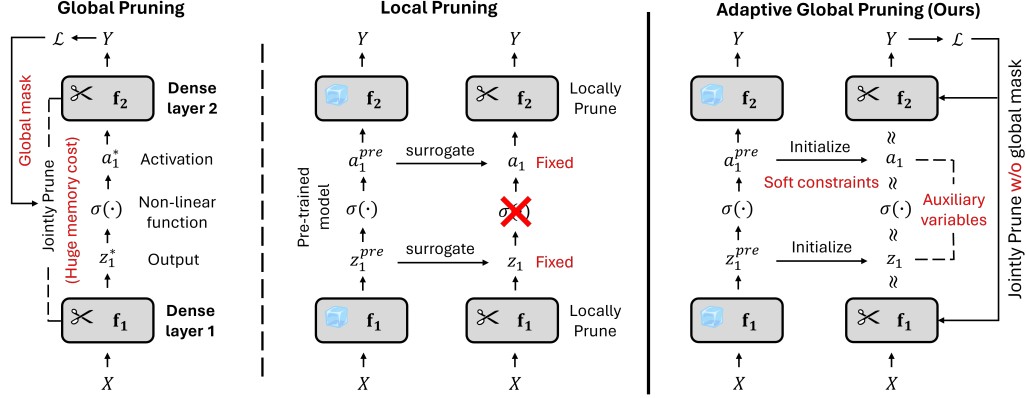

Figure 4: **Illustration of *SparseLLM* pruning method** compared to *conventional global pruning* and *local pruning*. We consider a *two-layer* neural network as an abstraction for simplicity. *Global pruning* (**left**) is memory prohibitive due to poor scalability. *Local pruning* (**mid**) considers pruning each layer independently, while inevitably sacrificing performance due to the ignorance of global supervision. Our adaptive global pruning (**right**) achieves global pruning with low memory cost by leveraging auxiliary variables and soft constraints.

In the middle, local pruning is shown, where each layer is pruned independently. This approach reduces memory costs by applying separate masks to each layer. However, it inevitably sacrifices performance because it ignores global supervision, which can lead to suboptimal pruning decisions that do not consider the network as a whole.

On the right, the adaptive global pruning method of *SparseLLM* is presented. This method achieves global pruning with low memory cost by leveraging auxiliary variables and soft constraints. It combines the benefits of global pruning—considering the entire network structure—with efficient memory usage. The introduction of auxiliary variables allows for flexible and adaptive pruning, ensuring that the overall performance of the network is maintained while keeping memory costs low.

Thus, the figure highlights the trade-offs between different pruning strategies. Conventional global pruning incurs high memory costs, local pruning reduces memory usage at the expense of performance, and the adaptive global pruning of *SparseLLM* strikes a balance by maintaining performance with lower memory requirements through the use of auxiliary variables and soft constraints.

### A.3 Calibration Samples

Figure 5 and Figure 6 present how perplexity changes with the calibration sample sizes on the datasets PTB and C4 for OPT-2.7b and LlaMA-2 7b, respectively. In both figures, as the number of calibration samples increases, the perplexity decreases for both SparseGPT and *SparseLLM*. This indicates that having more calibration samples can be beneficial in the pruning process. Perplexity decreases more rapidly from 8 samples to 32 samples. Beyond 32 samples, the rate at which perplexity decreases starts to slow down. In addition, increasing the number of calibration samples requires more computational resources, e.g., memory and computation time, in the overall pruning process. This suggests that the calibration sample sizes should be between 32 and 64 to ensure good performance while maintaining computational efficiency. Lastly, the figures show that *SparseLLM* achieves better perplexity than SparseGPT does with 32 or larger sample sizes for both OPT and LlaMA-2 models.

### A.4 Computation Time vs. Model Sizes

We study how the computation time per layer of SparseGPT and *SparseLLM* varies with different model sizes, as illustrated in Table 6 and Table 7 for OPT models and LlaMA-2 models. The rate at which the time taken increases is comparable for SparseGPT and *SparseLLM* as the model size increases. Additionally, computation time for *SparseLLM* are reported for a configuration of 4 to 10 epochs. As we have reported in Section 5, *SparseLLM* can reduce the training loss in as few as 2 to 3 epochs. This suggests that the proposed *SparseLLM* remains computationally efficient.

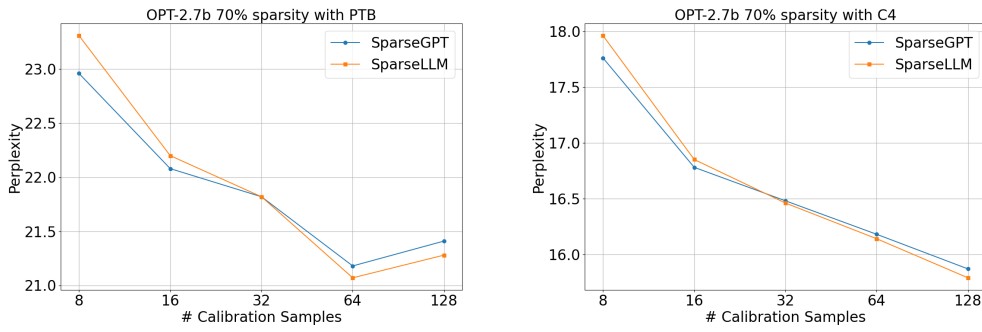

Figure 5: Sensitivity of OPT-2.7b on the calibration sample sizes for datasets PTB and C4.

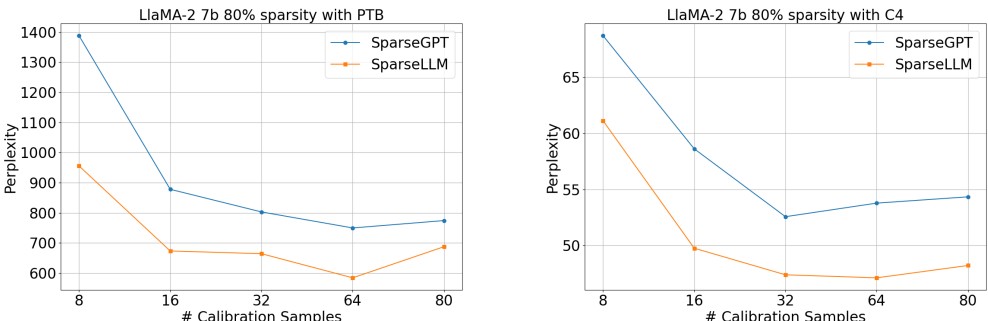

Figure 6: Sensitivity of LlaMA-2 7b models on the calibration sample sizes for datasets PTB and C4.

Table 6: Computation time in seconds of OPT models.

| Method | OPT-125m | OPT-1.3b | OPT-2.7b | OPT-6.7b | OPT-13b | OPT-30b | OPT-66b |
|---|---|---|---|---|---|---|---|
| SparseGPT | 2.30 | 10.18 | 18.35 | 24.40 | 28.65 | 48.91 | 103.19 |
| SparseLLM | 16.34 | 22.79 | 42.86 | 174.08 | 85.62 | 174.07 | 284.59 |

Table 7: Computation time in seconds of LlaMA-2 models.

| Method | Llama-2 7b | Llama-2 13b |
|---|---|---|
| SparseGPT | 11.94 | 16.58 |
| SparseLLM | 146.80 | 252.48 |

## A.5 Experiment Results for Additional Models

Detailed results on perplexity and zero-shot task accuracy for additional models are reported in Table 8 and Table 9. Similar to other models, we report the perplexity results for the dense model next to the name of the model in the table. In particular, we see that SparseGPT and *SparseLLM* outperform Magnitude and Wanda with a significant margin across different sparsity. *SparseLLM* shares similar perplexity with SparseGPT for smaller sparsity but demonstrates much better perplexity for larger sparsity. Similar perplexity trends are observed across all three datasets, although, PTB, having the highest perplexity for each sparsity and method, is likely the most challenging dataset among the three. For the zero-shot taks accuracy, we see that *SparseLLM* achieves comparable results to SparseGPT for smaller perplexity and the performance improvements are more obvious and significant with higher sparsity.

Table 8: Perplexity in high sparsity regimes ($\geq 70\%$); the lower the perplexity, the better.

| OPT-125m (WikiText2 (WT2): 27.66; PTB: 38.99; C4: 26.56) | | | | | | | | | | | |
|---|---|---|---|---|---|---|---|---|---|---|---|
| Sparsity | 70% | | | 80% | | | 90% | | | 3:4 | | |
| Dataset | WT2 | PTB | C4 | WT2 | PTB | C4 | WT2 | PTB | C4 | WT2 | PTB | C4 |
| Magnitude | 3806.96 | 3429.35 | 2263.37 | 4890.96 | 4121.49 | 3213.85 | 6613.18 | 5380.80 | 4475.29 | - | - | - |
| Wanda | 351.83 | 412.52 | 248.94 | 1912.45 | 2512.93 | 1066.86 | 4940.89 | 4337.27 | 3126.02 | - | - | - |
| SparseGPT | 239.26 | 265.83 | 156.33 | 2072.12 | 1952.85 | 1050.83 | 6131.57 | 6963.27 | 2443.33 | 1482.61 | 2215.44 | 657.26 |
| SparseLLM | 208.46 | 255.75 | 137.72 | 1358.10 | 1418.09 | 654.54 | 5291.64 | 5067.41 | 2003.09 | 914.87 | 1210.84 | 450.01 |

| OPT-6.7b (WikiText2 (WT2): 10.86; PTB: 15.77; C4: 12.71) | | | | | | | | | | | |
|---|---|---|---|---|---|---|---|---|---|---|---|---|
| Sparsity | 70% | | | 80% | | | 90% | | | 3:4 | | |
| Dataset | WT2 | PTB | C4 | WT2 | PTB | C4 | WT2 | PTB | C4 | WT2 | PTB | C4 |
| Magnitude | 7054.21 | 5437.44 | 4850.25 | 7937.49 | 5971.86 | 6031.54 | 2.4e4 | 2.5e4 | 2.1e4 | - | - | - |
| Wanda | 54.95 | 129.73 | 116.67 | 1493.58 | 1196.93 | 996.00 | 2.1e4 | 2.0e4 | 1.8e4 | - | - | - |
| SparseGPT | 12.27 | 18.90 | 15.28 | 31.04 | 51.26 | 29.42 | 8871.24 | 5713.57 | 3797.20 | 570.08 | 361.81 | 328.18 |
| SparseLLM | 12.16 | 18.39 | 14.93 | 23.96 | 39.32 | 26.97 | 2095.85 | 1842.48 | 953.44 | 83.36 | 128.99 | 62.11 |

Table 9: Accuracy (%) of zero-shot tasks; the higher the accuracy, the better.

| OPT-6.7b | | | | | | | | | |
|---|---|---|---|---|---|---|---|---|---|
| Sparsity | Method | BoolQ | RTE | HellaSwag | WinoGrande | ARC-e | ARC-c | OBQA | Mean |
| Dense | | 66.12 | 56.03 | 50.49 | 65.27 | 65.72 | 30.63 | 27.60 | 51.69 |
| 70% | SparseGPT | 61.74 | 54.87 | 48.46 | 63.85 | 64.31 | 29.27 | 25.40 | 49.70 |
| | SparseLLM | 60.61 | 54.51 | 48.8 | 62.9 | 64.14 | 30.03 | 26.60 | 49.66 |
| 80% | SparseGPT | 55.08 | 48.38 | 42.22 | 59.43 | 57.79 | 25.85 | 21.40 | 44.31 |
| | SparseLLM | 58.69 | 51.26 | 43.78 | 59.67 | 58.38 | 26.88 | 22.00 | 45.81 |
| 90% | SparseGPT | 38.53 | 53.07 | 26.00 | 48.07 | 26.81 | 21.67 | 14.40 | 32.65 |
| | SparseLLM | 46.48 | 52.71 | 26.21 | 51.70 | 27.44 | 19.71 | 13.40 | 33.95 |
| 3:4 | SparseGPT | 46.70 | 54.15 | 28.82 | 51.07 | 32.45 | 18.17 | 15.40 | 35.25 |
| | SparseLLM | 53.49 | 53.42 | 36.24 | 53.51 | 43.94 | 22.61 | 17.40 | 40.09 |

## A.6 Hyperparameter $\alpha$ and $\beta$ Selection

Hyperparameters $\alpha$ and $\beta$ are used in Eq. 5. We select $\alpha$ and $\beta$ from the set $\{0.01, 0.1, 1, 5, 10, 100\}$ and perform a study on models to understand the impact of the hyperparameters. Results for OPT-1.3b with 70% sparsity is shown in Table 10.

Table 10: Ablations of the hyperparameters $\alpha$ and $\beta$ on OPT-1.3b with 70% sparsity (in perplexity)

| $\alpha$ / $\beta$ | 0.01 | 0.1 | 1 | 5 | 10 | 100 |
|---|---|---|---|---|---|---|
| 0.01 | 18.01 | 17.97 | 17.97 | - | - | - |
| 0.1 | 18.04 | **17.82** | 17.96 | 18.04 | 18.40 | - |
| 1 | 18.20 | 18.02 | 18.11 | 17.87 | 17.96 | 18.22 |
| 5 | 18.06 | 18.02 | 18.03 | 17.92 | 17.96 | 18.04 |
| 10 | 18.03 | 18.01 | 17.96 | 17.96 | 17.96 | 18.03 |
| 100 | 18.04 | 18.04 | 17.98 | 18.01 | 18.01 | 18.03 |

## A.7 Limitations and Future Work

While *SparseLLM* marks a significant step forward in the efficient pruning of large language models, it is important to acknowledge the inherent trade-offs associated with any model compression technique. Firstly, while our method reduces the complexity of LLMs and enhances computational efficiency, there is an inevitable balance between sparsity and performance that requires careful calibration. Additionally, in this work, we still assume homogeneous sparsity, i.e., the pruning sparsity for each layer is the same and equal to the global sparsity. How to achieve heterogeneous sparsity under our framework and fully fulfill the potential of global pruning is of great interest. Lastly, the effectiveness of *SparseLLM*, like any pruning method, may vary across different models and tasks, and its generalizability to all scenarios remains an area for further exploration.

