# OpenReview forum: "SparseLLM: Towards Global Pruning of Pre-trained Language Models"
_NeurIPS.cc/2024/Conference — NeurIPS 2024 poster_

### Official Review · Reviewer_K3H3 · 2024-07-12

**Soundness:** 2
**Presentation:** 2
**Contribution:** 2
**Rating:** 5
**Confidence:** 2

**Summary:**

This paper tries to improve the pruning technique for LLM to enhance computational and memory efficiency. The proposed SparseLLM, circumvents the scalability problem of global pruning and suboptimal performance due to local pruning. It breaks down global pruning into subproblems

**Strengths:**

1. Pruning LLM remains to be an interesting problem given the size of LLM keeps growing and serving these models on less powerful devices are worth researching.
2. The idea of decomposing the global pruning into smaller subproblems and conceptualizing LLM as several modular functions are practical.

**Weaknesses:**

1. It appears the pruning procedure differs from model to model. If the model architecture changes, the pruning needs to be adjusted. Therefore, I have doubts on the generality of the proposed solution.
2. The costs of pruning are not specified in the evaluation. For example , in Table 2, it seems for most of the tasks, SparseLLM has a comparable performance as SparseGPT. In this case, what is the advantage of SparseLLM?

**Questions:**

1. Regarding the global pruning, what is it necessary to fit the models within one GPU? Why not applying tensor parallel?
2. What is the comparison of training loss convergence between SparseLLM and other baseline methods?

**Limitations:**

The authors address the limitations and there is no potential negative societal impact of this work.

---

> ### Author Rebuttal · Authors · 2024-08-07
>
> Dear Reviewer K3H3,
>
> Thank you for finding our method interesting and practically useful. Please refer to our response below for details:
>
> > *"It appears the pruning procedure differs from model to model. If the model architecture changes, the pruning needs to be adjusted. Therefore, I have doubts about the generality of the proposed solution."*
>
> **A1.**
> The pruning procedure such as the closed-form solutions for each subproblem in our *SparseLLM* could be different from model to model, but the generality of our method roots in the mathematical formulation of our Eq. 3 and is guaranteed theoretically. More specifically, as long as the neural network architecture satisfies the Directed Acyclic Graph (DAG), which is, in general, the case for LLMs, our Eq. 3 of *SpasreLLM* can handle that.
>
> That being said, we have discussed this weakness in the limitation section of our manuscript. We are happy to include more discussion there and explore extending *SparseLLM* onto more diverse model architectures in the future work.
>
>
> > *"The costs of pruning are not specified in the evaluation. For example, in Table 2, it seems for most of the tasks, SparseLLM has a comparable performance to SparseGPT. In this case, what is the advantage of SparseLLM?"*
>
> **A2.**
> *SparseLLM* consistently achieves competitive results, significantly decreasing perplexity by up to 80% compared to SparseGPT and Wanda. Notable improvements include OPT-2.7b at 90% sparsity: *SparseLLM* achieves a perplexity of 759.11 versus SparseGPT's 2285.01 for PTB, and 527.70 versus 1776.08 for C4, representing over **60%** improvements in both cases. For OPT-125m at 80% sparsity for C4, *SparseLLM* achieves a perplexity of 654.54 versus SparseGPT's 1050.83, representing over **40%** improvement.
>
> *SparseLLM* is a generic framework, that both local pruning and global pruning are special cases. By flexibly switching between these extremes, the computational complexity of *SparseLLM* can be the same as local pruning.
>
> > *"Regarding the global pruning, what is it necessary to fit the models within one GPU? Why not apply tensor parallel?"*
>
> **A3.**
> In this work, we consider pruning methods for LLMs under resource-constrained environments, where it is likely that only one GPU is available. Representative examples include academic labs and hospitals, edge devices, and mobile computing, which are prevalent in reality.
>
> By utilizing more computational resources (e.g., bigger and more powerful GPUs, distributed training including tensor parallel, pipeline parallel, etc) one can to some extent achieve the vanilla or extreme global pruning of LLMs, but that goes beyond the main focus of this work. However, we agree how to achieve the extreme global pruning from a high-performance computing perspective is definitely a challenging but interesting future direction on the basis of our manuscript.
>
> > *"What is the comparison of training loss convergence between SparseLLM and other baseline methods?"*
>
> **A4.**
> Since baseline methods such as SparseGPT and Wanda both consider one-shot pruning, there is no convergence curve for those methods. However, it is still possible to compare the training loss of *SparseLLM* over SparseGPT and Wanda. For example, under the setting of pruning layer 3 of OPT-125M at 80% sparsity, the training loss of our *SparseLLM* is **0.3** after 2 epochs and is **0.15** when convergence is achieved. On the contrary, the training loss of SparseGPT is **0.8**, after the one-shot pruning. Hence, *SparseLLM* can further reduce the training loss of one-shot pruning methods such as SparseGPT and Wanda via an iterative algorithm.

---

> > ### Comment · Reviewer_K3H3 · 2024-08-14
> >
> > Thanks for the response. I appreciate the authors' effort!

---

> > > ### Author Response · Authors · 2024-08-14
> > >
> > > Dear Reviewer K3H3,
> > >
> > > You are welcome :)
> > >
> > > We greatly appreciate your constructive comments, as well as the time and patience you have dedicated to this review.
> > >
> > > Best regards,
> > >
> > > Authors

---

### Official Review · Reviewer_35o1 · 2024-07-13

**Soundness:** 3
**Presentation:** 3
**Contribution:** 3
**Rating:** 5
**Confidence:** 3

**Summary:**

In this paper, the author proposes a method to globally prune large language models (LLMs) without consuming significant memory. By using auxiliary variables, the LLM can be pruned separately while maintaining dependencies. The evaluation demonstrates that the proposed method outperforms previous approaches in terms of perplexity and accuracy across various sparsity settings.

**Strengths:**

1. The proposed method is promising and innovative.
2. The paper is well-written and easy to follow.
3. The experiments are comprehensive, with detailed settings provided.

**Weaknesses:**

1. The use cases appear tricky regarding model size. For smaller models (<7B), they can fit into GPU memory (A100), allowing global pruning. For larger models (>70B), achieving 90% sparsity does not outperform the smaller versions (7B).
2. The work only considers unstructured pruning. Unstructured sparsity may not accurately reflect the actual model size and inference time.
3. There is no discussion on memory consumption across various model sizes and pruning methods, which may undermine the claim that global pruning is infeasible for LLMs due to memory constraints.
4. Although it outperforms previous methods, the performance at 90% sparsity drops significantly, reducing its practical usefulness.

**Questions:**

1. What model sizes benefit more from the proposed pruning method?
2. Can this method be extended to structured pruning, which may be more useful?

**Limitations:**

Please refer to the weaknesses section.

---

> ### Author Rebuttal · Authors · 2024-08-07
>
> Dear Reviewer 35o1,
>
> We are grateful for your recognition of the novelty of our method. Please find our detailed response below:
>
> > *"The use cases appear tricky regarding model size. For smaller models (<7B), they can fit into GPU memory (A100), allowing global pruning. For larger models (>70B), achieving 90% sparsity does not outperform the smaller versions (7B)."*
>
> **A1.**
> In this work, our major goal is to improve local pruning methods for LLMs under **resource-constrained** environments, where high-end GPUs such as A100 are typically unavailable. Representative examples include academic labs and hospitals, edge devices, and mobile computing, which are very prevalent in reality.
>
> Globally pruning LLMs with sizes greater than or equal to 7B under resource constraints is typically infeasible and might require distributed training. The major contribution of *SparseLLM* is the systematic exploration of global and local (layer-wise) pruning and everything in between using a theoretically sound technique. This allows us to decompose global pruning into smaller and decoupled subproblems that can be seamlessly combined with distributed and resource-efficient training.
>
> Moreover, *SparseLLM* can provide larger pruned models that outperform smaller dense models. For instance, we tested *SparseLLM* with 40% sparsity on Llama-2-13B and achieved a perplexity of **5.09** on WT2 and **7.01** on C4, which are **lower** than dense Llama-2-7B (5.47 on WT2 and 7.26 on C4). This shows that our approach is getting close to Pareto optimality and just needs a little push.
>
> > *"The work only considers unstructured pruning. Unstructured sparsity may not accurately reflect the actual model size and inference time."*
>
> **A2.**
> In this work, we considered not only unstructured pruning but also $N$:$M$ sparsity, or **semi-structured** pruning. The $N$:$M$ sparsity pruning requires that every $M$ consecutive parameters have at least $N$ zero elements. This can leverage NVIDIA’s sparse tensor cores to accelerate matrix multiplication in practice [1] *Asit Mishra, et al. "Accelerating sparse deep neural networks." arXiv preprint arXiv:2104.08378, 2021*. As shown in Table 1 (of the original manuscript), *SparseLLM* achieves competitive performance with 3:4 sparsity pruning in most cases, indicating its potential for actual GPU acceleration.
>
> > *"There is no discussion on memory consumption across various model sizes and pruning methods, which may undermine the claim that global pruning is infeasible for LLMs due to memory constraints."*
>
> **A3.**
> We provide the memory cost analysis of global pruning as below:
>
> For zero-order pruning methods, such as magnitude pruning, the memory cost is relatively low because there is no need for forward or backward propagation. The weights are pruned based solely on their absolute values, resulting in a memory complexity of $O(N)$, where $N$ is the size of the LLM model.
>
> For first-order methods that use derivatives to estimate the score of each parameter for pruning, an end-to-end forward and backward propagation is required to estimate the pruning mask. Additionally, adjusting the unpruned weights also necessitates another end-to-end propagation. The memory complexity for such methods includes storing the parameters, optimizer states, activations, gradients, and pruning mask, leading to a total memory cost (in GB) of $5$-$10\times$ model size, which is already prohibitive for most LLMs on a normal GPU.
>
> For second-order methods that use the Hessian, which are the most effective and commonly-used pruning methods, the memory cost is significantly higher due to the need to store and compute second-order derivatives. This results in a memory complexity of $O(N^2)$.
>
> In conclusion, global pruning with first or second-order methods is extremely memory expensive, making them impractical for large-scale LLMs. Our proposed method, *SparseLLM*, is very useful as it decomposes the global pruning problem into manageable subproblems, significantly reducing the memory requirements and making it feasible for resource-constrained environments.
>
> > *"Although it outperforms previous methods, the performance at 90\% sparsity drops significantly, reducing its practical usefulness."*
>
> **A4.**
> The 90% sparsity level is just one of the sparsities we have shown in our experiments. Despite the significant challenge, the performance improvement of our method over baselines at 90% sparsity is meaningful, demonstrating that *SparseLLM* can achieve better perplexity than SparseGPT and Wanda can even at extremely high sparsity levels, which is a non-trivial achievement.
>
> We believe the significant performance improvement of *SparseLLM* remains useful, as it can provide a better initialization for the "prune and re-train" approach. Re-training methods such as [2] *Zhang, Yuxin, et al. "Dynamic Sparse No Training: Training-Free Fine-tuning for Sparse LLMs." ICLR 2024* could complement our *SparseLLM*, potentially close the gap and lead to more practically useful models.
>
> > *"What model sizes benefit more from the proposed pruning method?"*
>
> **A5.**
> We empirically proved that *SparseLLM* can achieve competitive performance over comparison methods over various model sizes, from 125 million up to 66 billion. The performance gap between *SparseLLM* and other methods at a given sparsity will in general decrease as the model size increases, which, however, is due to that larger models are more over-parameterized and easier to prune. This monotonic decreasing pattern is true for all pruning methods including SparseGPT and Wanda.
>
> > *"Can this method be extended to structured pruning, which may be more useful?"*
>
> **A6.**
> Although in this work we mainly focus on unstructured and $N$:$M$ sparsity pruning, we believe extending our *SparseLLM* further to structured pruning is potentially feasible and interesting to explore. The high-level idea of *SparseLLM* is generic and not restricted to specific model pruning algorithms.

---

### Official Review · Reviewer_snMo · 2024-07-13

**Soundness:** 2
**Presentation:** 4
**Contribution:** 2
**Rating:** 5
**Confidence:** 4

**Summary:**

This paper introduces SparseLLM, a novel pruning technique targeted at the FFN layers in LLMs. By treating global and local (layer-wise) pruning as special cases in the proposed formulation, SparseLLM can circumvent the limitations of both extremes. The proposed method introduces auxiliary variables and soft constraints within LLM feedforward layers, which helps decompose pruning into subproblems that can be solved analytically. SparseLLM is evaluated on modern LLMs such as OPT and LLaMa2, pruning their weights in both unstructured and N:M patterns; the performance is compared with other state-of-the-art LLM pruning techniques such as SparseGPT and Wanda.

**Strengths:**

* The authors address an important and relevant problem: how to induce relatively high unstructured and semi-structured sparsity (>70%) into LLMs both cheaply and effectively (in terms of accuracy w.r.t. dense model).
* The paper is very well-written and the core ideas have been presented clearly in an easy-to-understand manner.
* The proposed formulation permits the systematic exploration of global and local (layer-wise) sparsity and everything in between. This is quite powerful.
* The benchmarks are fairly strong, using modern medium-scale language models and comparisons to SoTA approaches like SparseGPT/Wanda; however, performance is a bit lacking (see comments below).

**Weaknesses:**

* The proposed approach appears to obtain accuracy figures on par with SparseGPT at 70% unstructured sparsity. In higher sparsity regimes, SparseLLM seems to outperform SparseGPT and Wanda; however, most of the perplexity values reported for these regimes by all approaches (including SparseLLM) are extremely high; I'd argue that the obtained zero-shot models are pretty much unusable in the real world. I'm not sure I understand why saving pruning+retraining time is important in this regime - wouldn't additional training to close this gap make more sense? If so, SparseLLM could potentially be a good initialization for such an approach.
* Why are there no experiments performed for 2:4 sparsity, especially since results for 3:4 are reported? 2:4 is particularly relevant since hardware acceleration for this sparsity pattern is possible with today's GPUs. Does SparseLLM outperform SparseGPT/Wanda in this case?

**Questions:**

* To clarify, does 3:4 mean that 3 out of 4 consecutive elements are zero? N:M sparsity is traditionally defined differently: at most N out of M consecutive elements are non-zero.
* Can SparseLLM handle alternative activation functions such as GeLU?
* Can SparseLLM handle networks with residual connections between layers?

**Limitations:**

Limitations are adequately discussed in Section A.7 (Appendix).

---

> ### Author Rebuttal · Authors · 2024-08-07
>
> Dear Reviewer snMo,
>
> We sincerely appreciate that you found our paper and method interesting with solid results. Please refer to our response below for details:
>
> >  *"The proposed approach appears to obtain accuracy figures on par with SparseGPT at 70% unstructured sparsity. In higher sparsity regimes, SparseLLM seems to outperform SparseGPT and Wanda; however, most of the perplexity values reported for these regimes by all approaches (including SparseLLM) are extremely high; I'd argue that the obtained zero-shot models are pretty much unusable in the real world. I'm not sure I understand why saving ``pruning and retraining" time is important in this regime - wouldn't additional training to close this gap make more sense? If so, SparseLLM could potentially be a good initialization for such an approach."*
>
> **A1.**
> *SparseLLM* achieves an accuracy improvement over SparseGPT by an average of 1-3% on zero-shot tasks. For example, at 70% sparsity with LLaMA-2 7b on the WinoGrande dataset, *SparseLLM* outperforms SparseGPT by 2.3 percentage points (61.39 vs. 59.04), and at 70% sparsity with LLaMA-2 13b on the RTE dataset, *SparseLLM* shows a 3.25 percentage point improvement (61.73 vs. 58.48).
>
> In higher sparsity regimes, for example, 80% sparsity, the perplexity of *SparseLLM* on datasets WT2 and C4 is practically useful. For instance, *SparseLLM* achieves perplexity values of 15.61 and 16.61 on WT2 and C4 for OPT-30b at 80% sparsity. Additionally, *SparseLLM* achieves perplexity values of 16.45 and 17.70 on WT2 and C4 for OPT-66b at 80% sparsity. The perplexity of pruned model by *SparseLLM* is close to that of some smaller dense models, and just needs a little push.
>
> *SparseLLM* could provide a better initialization for the "prune and re-train" approach, potentially enhancing performance and closing the gap caused by pruning. Re-training methods such as [1] *Zhang, Yuxin, et al. "Dynamic Sparse No Training: Training-Free Fine-tuning for Sparse LLMs." ICLR 2024* could complement our *SparseLLM*. We will explore such approaches and discuss this topic in the limitations and future work section of our paper.
>
> > *"Why are there no experiments performed for 2:4 sparsity, especially since results for 3:4 are reported? 2:4 is particularly relevant since hardware acceleration for this sparsity pattern is possible with today's GPUs. Does SparseLLM outperform SparseGPT/Wanda in this case?"*
>
> **A2.**
> We have added the results for 2:4 sparsity in Table 5 in our one-page PDF, where *SparseLLM* can consistently beat the comparison methods, which demonstrates the potential of *SparseLLM* to achieve practical GPU acceleration.
>
> > *"To clarify, does 3:4 mean that 3 out of 4 consecutive elements are zero? N:M sparsity is traditionally defined differently: at most N out of M consecutive elements are non-zero."*
>
> **A3.**
> In our paper, $N$:$M$ means every group of consecutive $M$ values contains at least $N$ zeros, which we follow the definition from [2] *Mishra, Asit, et al. "Accelerating sparse deep neural networks." arXiv preprint arXiv:2104.08378 (2021).*
>
> > *"Can SparseLLM handle alternative activation functions such as GeLU?"*
>
> **A4.**
> Yes, *SparseLLM* can handle alternative activation functions such as GeLU in a similar way to how it handles SiLU. Both activation functions are element-wise operators, meaning each output position is calculated independently. By following the approach used for the SiLU activation function in LLaMA (Lines 233-238), leveraging a pre-computed look-up table, one can obtain analytical solutions for each subproblem.
>
> > *"Can SparseLLM handle networks with residual connections between layers?"*
>
> **A5.**
> Yes, *SparseLLM* can handle architectures with residual connections. An intuitive way to explain this is by referring to Figure 2 in our original manuscript. Specifically, in the sub-figure of *SparseLLM* on the LLaMA layer, residual connections can be regarded as a special case of the bottom half of the FFN module in the LLaMA layer. In this case, the "up proj" layer is replaced by a trivial identity mapping, and the dot-product aggregation (black round dot) is replaced by summation. Since the "up proj" layer is replaced by a non-parametric function, the auxiliary variable $z_{\ell}$ can be discarded.

---

> ### Author Response · Authors · 2024-08-12
> **Request to review the rebuttal [Author-Reviewer discussion phase ending soon]**
>
> Dear Reviewer snMo,
>
> We would like to sincerely thank you again for your valuable feedback and insights, which have greatly improved our paper. We promise to reflect all your comments in the final manuscript thoroughly. As we are towards the end of the author-reviewer discussion period, we kindly request you to please go through our rebuttal, and we would be immensely grateful if you could reconsider your recommendation.
>
> Best regards,
>
> Authors

---

> > ### Comment · Reviewer_snMo · 2024-08-12
> >
> > Thank you for the response. I appreciate the new 2:4 results (among others) given the limited rebuttal timeframe. I will raise my score.

---

> > > ### Author Response · Authors · 2024-08-12
> > >
> > > Dear Reviewer snMo,
> > >
> > > Thank you very much for acknowledging our responses. We are pleased to see that you have raised your score.
> > >
> > > We greatly appreciate your constructive comments, as well as the time and patience you have dedicated to this review.
> > >
> > > Best regards,
> > >
> > > Authors

---

### Official Review · Reviewer_FMFD · 2024-07-15

**Soundness:** 3
**Presentation:** 3
**Contribution:** 3
**Rating:** 5
**Confidence:** 4

**Summary:**

This paper presents SparseLLM, a framework to prune large language models by decomposing the global pruning objective into multiple sub-problems, each of which can be solved with low resources, when combined, solve the global pruning objective. The method reformulates LLMs as a chain of modular functions and uses auxiliary variables to enable problem decomposition. Empirically, SparseLLM shows consistent improvements over the local pruning methods especially in high-sparsity regimens.

**Strengths:**

1. The paper introduces a novel method to address the limitations of both global and local pruning for large language models.
2. The proposed approach is well-grounded in theory with clear mathematical foundations.

**Weaknesses:**

1. The experiments are a bit weak in model choice. Older models like OPT and Llama-2 are chosen, when a lot of new and better performing models have come out like Mistral, Gemma, and even Llama 3 (which came out in April).
2. Experiments based on lower sparsity levels are also missing, I would like to see the comparison and time computations at 10/20/50% sparsity as well.
3. The dense baseline (0% sparsity) is missing from all tables. It's important to gauge the effectiveness of the proposed method.
4. For small models, it seems that SparseLLM performs on par with SparseGPT, with the added computational complexity.
5. Ablations on the effectiveness of the hyperparameters $\alpha$ and $\beta$ are missing.

**Questions:**

I've asked most of my questions in the weaknesses section above.

**Limitations:**

The limitations section is a bit vague and only discusses the requirement for certain structural properties of the network to be satisfied in order to demonstrate the effectiveness of SparseLLM. I would like to add that this does not affect my score.

---

> ### Author Rebuttal · Authors · 2024-08-07
>
> Dear Reviewer FMFD,
>
> We sincerely appreciate that you found our paper and method interesting with solid results. Please refer to our response below for details:
>
> > *"The experiments are a bit weak in model choice. Older models like OPT and Llama-2 are chosen, when many new and better-performing models have come out like Mistral, Gemma, and even Llama 3 (which came out in April)."*
>
> **A1.**
> We have performed additional experiments on the LlaMA-3 8b model and presented the perplexity results in Table 1 of our one-page PDF. Our results demonstrate that *SparseLLM* achieves competitive performance on the latest LlaMA-3 model, confirming its applicability to state-of-the-art models.
>
> > *"Experiments based on lower sparsity levels are also missing, I would like to see the comparison and time computations at 10/20/50% sparsity as well."*
>
> **A2.**
> We provide the perplexity and computation time results for sparsity 10/20/50% on OPT-1.3b and OPT-2.7b in Table 2 and Table 3 in our one-page PDF.  We see similar perplexity results for all four methods for the 10% and 20% sparsity, as naive magnitude pruning can achieve pretty good perplexity. However, we start to see improvements in perplexity from *SparseLLM* starting from 50% sparsity and the improvements are more significant with subsequent even higher sparsity, as shown in the original manuscript.
>
> > *"The dense baseline (0% sparsity) is missing from all tables. It's important to gauge the effectiveness of the proposed method."*
>
> **A3.**
> The performance of the dense baselines was provided in our original manuscript. In Table 1 (of the original manuscript), it is placed in parentheses after each LLM's name. In Table 2 (of the original manuscript), it is marked as "Dense" below the row of dataset names. We will modify our paper to improve its readability in the future.
>
> > *"For small models, it seems that SparseLLM performs on par with SparseGPT, with the added computational complexity."*
>
> **A4.**
> *SparseLLM* consistently achieves competitive results for small models (e.g., OPT-125m, OPT-1.3b, and OPT-2.7b), significantly decreasing perplexity by up to 80% compared to SparseGPT and Wanda. Notable improvements include OPT-2.7b at 90% sparsity: *SparseLLM* achieves a perplexity of 759.11 versus SparseGPT's 2285.01 for PTB, and 527.70 versus 1776.08 for C4, representing over **60%** improvements in both cases. For OPT-125m at 80\% sparsity for C4, *SparseLLM* achieves a perplexity of 654.54 versus SparseGPT's 1050.83, representing over **40%** improvement.
>
> *SparseLLM* is a generic framework, with mathematical proof that both local pruning and global pruning are special cases. By flexibly switching between these extremes, the computational complexity of *SparseLLM* can be the same as local pruning.
>
> > *"Ablations on the effectiveness of the hyperparameters $\alpha$ and $\beta$ are missing."*
>
> **A5.**
> We present the ablation studies for the hyperparameters $\alpha$ and $\beta$ in Table 4 of our one-page PDF. Specifically, we consider the model OPT-1.3b with 70% sparsity on the WikiText2 dataset. We vary the values of $\alpha$ and $\beta$ from the set {$0.01, 0.1, 1, 5, 10, 100$} and compare the resulting perplexity. In the table, we use ``-" to indicate instances of numerical instability. The best combination of $\alpha$ and $\beta$ we found is $(0.1, 0.1)$, while the perplexity is in general insensitive to the choice of $\alpha$ and $\beta$, which is a good sign, meaning that our method is robust to the choice of hyperparameters.

---

### Author Rebuttal · Authors · 2024-08-07

Dear Reviewers,

We sincerely thank all your professional and constructive comments, especially given the time and workload for this year's NeurIPS. We have provided detailed responses to each individual comment and hope we have addressed all your concerns. Below is a brief summary of the new experimental results added during the rebuttal period:

- Results for LlaMA-3 8B
- Low sparsity regime results: 10%, 20%, and 50% sparsity for all pruning methods
- Ablation study on the hyperparameters $\alpha$ and $\beta$
- 2:4 sparsity results

**All newly added results have been compiled into our one-page PDF, which we invite you to review for further details.**

Sincerely,
The Authors

---

### Decision · Program_Chairs · 2024-09-25

**Decision:**

Accept (poster)

**Comment:**

The paper introduces SparseLLM, a framework designed to globally prune large language models (LLMs) by decomposing the pruning process into manageable subproblems. This approach allows for resource-efficient optimization and demonstrates significant performance improvements, particularly in high-sparsity regimes where it surpasses current state-of-the-art methods. SparseLLM conceptualizes LLMs as a chain of modular functions and leverages auxiliary variables for problem decomposition, facilitating practical application on LLMs.

Reviewers agree the proposed method is noval, promising and theoretical grounding.  However, they also raised concerns including model choice in experiments, lack of lower sparsity level reults, and missing dense baselines.

The authors address reviewer concerns by providing additional experimental results, including those for the latest LlaMA-3 model, lower sparsity regime results, and ablation studies on hyperparameters. The rebuttals are structured, detailed and addressed most of the concerns.

All the reviewers give the same 5 ratings (weak accept).